# Quantum fluctuations lead to glassy electron dynamics in the good metal regime of electron doped KTaO₃

Shashank Kumar Ojha [1] ✉, Sankalpa Hazra[1,2,6], Surajit Bera[1,6], Sanat Kumar Gogoi [1,3], Prithwijit Mandal [1], Jyotirmay Maity[1], Andrei Gloskovskii[4], Christoph Schlueter [4], Smarajit Karmakar [5], Manish Jain [1], Sumilan Banerjee [1] ✉, Venkatraman Gopalan[2] & Srimanta Middey [1] ✉

One of the central challenges in condensed matter physics is to comprehend systems that have strong disorder and strong interactions. In the strongly localized regime, their subtle competition leads to glassy electron dynamics which ceases to exist well before the insulator-to-metal transition is approached as a function of doping. Here, we report on the discovery of glassy electron dynamics deep inside the good metal regime of an electron-doped quantum paraelectric system: KTaO₃. We reveal that upon excitation of electrons from defect states to the conduction band, the excess injected carriers in the conduction band relax in a stretched exponential manner with a large relaxation time, and the system evinces simple aging phenomena—a telltale sign of glassy dynamics. Most significantly, we observe a critical slowing down of carrier dynamics below 35 K, concomitant with the onset of quantum paraelectricity in the undoped KTaO₃. Our combined investigation using second harmonic generation technique, density functional theory and phenomenological modeling demonstrates quantum fluctuation-stabilized soft polar modes as the impetus for the glassy behavior. This study addresses one of the most fundamental questions regarding the potential promotion of glassiness by quantum fluctuations and opens a route for exploring glassy dynamics of electrons in a well-delocalized regime.

The notion of glassy dynamics associated with the electronic degree of freedom in condensed matter systems was first envisaged by Davies, Lee, and Rice in 1982[1]. Building heavily on Anderson's seminal work on the localization of wave functions in random disordered media[2], it was predicted that, in a disordered insulator with highly localized electronic states, the interplay between disorder and long-range Coulomb interaction (Fig. a) should precipitate electronic frustration in real space. Such a scenario would result in a rugged energy landscape with numerous metastable states, leading to the emergence of electron glass[3–5]. This hypothesis was soon tested on various strongly localized electronic systems, including granular metals, crystalline and amorphous oxides, and later, it was also extended to doped semiconductors and two-dimensional electron gases[3–7]. The typical relaxation time in these systems ranges from a few seconds to several hours, making

[1]Department of Physics, Indian Institute of Science, Bengaluru 560012, India. [2]Department of Materials Science and Engineering, The Pennsylvania State University, University Park, PA 16802, USA. [3]Department of Physics, Digboi College, Digboi 786171, India. [4]Deutsches Elektronen-Synchrotron DESY, 22607 Hamburg, Germany. [5]Tata Institute of Fundamental Research, 36/P, Gopanpally Village, Serilingampally Mandal, Ranga Reddy District, Hyderabad 500107, India. [6]These authors contributed equally: Sankalpa Hazra, Surajit Bera. ✉e-mail: shashank@iisc.ac.in; sumilan@iisc.ac.in; smiddey@iisc.ac.in

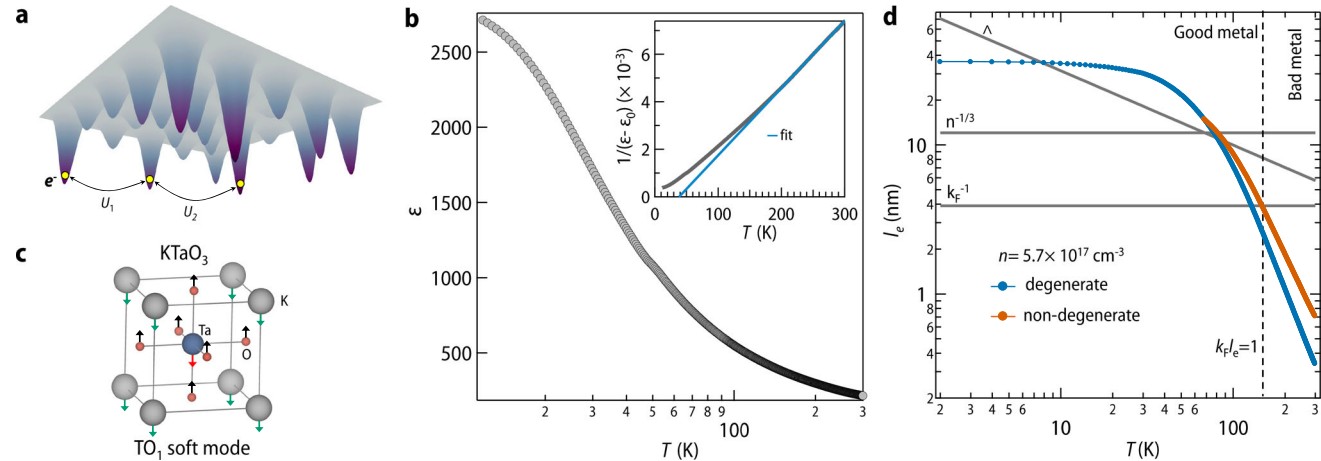

**Fig. 1 | Electron glass, quantum paraelectricity, and temperature dependence of electron's mean free path in KTaO$_{3-\delta}$. a** A schematic to describe the concept of electron glass in a disordered insulator with highly localized electronic states. Here the solid colored cones represent the random disorder potential and yellow-filled circles represent electrons trapped in them. Disorder tries to make the distribution of doped carriers random, however long-range Coulomb interactions ($U_1$, $U_2$) try to make the distribution homogeneous leading to electronic frustration. **b** Temperature-dependent dielectric constant ($\varepsilon$) of pristine (001) oriented KTaO$_3$ single crystal recorded at an AC frequency of 10 kHz, taken from our previous paper[16]. We further note that the value of $\varepsilon$ for our sample appears to be slightly lower than the reported values[60]. We attribute this difference to the difference in the sample preparation process, which may add slight oxygen vacancies, even in the pristine, as-received crystal[61]. Sitting on the verge of quantum critical point, KTaO$_3$ is long known for its peculiar dielectric properties[62] wherein the onset of

quantum fluctuation leads to a marked departure from classical paraelectric behavior below 35 K leading to saturation of $\varepsilon$ at low temperature. This is further evident from the modified Curie-Weiss fitting ($\varepsilon = \varepsilon_0 + \frac{C_W}{T-\theta_{CW}}$, $\varepsilon_0$ is temperature independent component, $C_W$ is Curie-Weiss constant and $\theta_{CW}$ is Curie-Weiss temperature, respectively) shown in the inset of panel (**b**). $\theta_{CW}$ obtained from the fitting is found to be 37 K. **c** Unit cell of pristine KTaO$_3$ along with the transverse optical soft mode. **d** Temperature-dependent electron's mean free path $l_e$ for oxygen-deficient KTaO$_3$ calculated within the Drude-Boltzmann picture. Blue and orange curves correspond to degenerate and non-degenerate cases, respectively. $\Lambda$ denotes the thermal de Broglie wavelength. A dotted vertical line marks the temperature above which $l_e$ becomes shorter than the inverse of Fermi wave-vector ($k_F$) and the sample crosses over from good metal to a bad metal phase. For details of the calculations, we refer to Supplementary Note 1 which significantly overlaps with our earlier work[15]. Source data are provided as a Source data file.

them an obvious choice for studying glassy physics in laboratory timescales. What makes these systems even more intriguing is the array of perturbations that one can use to effectively drive them away from equilibrium[4]. Furthermore, due to the light mass of electrons, electron glasses are highly susceptible to quantum fluctuations. This aspect introduces additional complexities in understanding the behavior of electron glasses[8].

In the antithetical regime of highly delocalized electrons i.e., in a metal, the screening effect significantly reduces the strength of the electron-electron and electron-impurity interactions. Consequently, such system generally possesses a non-degenerate ground state with a well-defined Fermi surface. As a result, the dynamics of glassy behavior, which involves the existence of multiple, competing ground states, is incompatible with the behavior of metals. In fact, the manifestation of glassiness fades away considerably prior to the transition from insulator to metal, and there is an absolute lack of any substantiated indication of the presence of glassiness within a good metal regime[3] where quasi-particle mean free path is larger than the electron's wavelength.

In this work, we report on the discovery of glassy dynamics of conduction electrons in an electron-doped quantum paraelectric system, namely KTaO$_3$, in a good metal regime. Even more surprising observation is that glassiness is found to appear in a regime where quantum fluctuations are inherently present in the system. In pristine KTaO$_3$, the quantum fluctuations associated with the zero point motion of the atoms prohibits the onset of ferroelectric order below 35 K (Fig. 1b), and the system's properties are typically governed by the presence of an associated low-energy transverse optical phonon (Fig. 1c) popularly known as soft polar mode[9]. Our combined transport and optical second harmonic generation measurements find that properties associated with the soft polar mode are preserved even deep inside the metallic regime. Most importantly, such soft modes are found to be directly responsible for emergent glassy dynamics at low temperatures, which is further corroborated by our theoretical calculations. Our observation is one of the rarest examples where quantum

fluctuation, which is generally considered as a bottleneck for electron glass formation[8], is ultimately accountable for the appearance of glassy dynamics in a good metallic phase.

## Results

### Demonstration of good metallic behavior

Due to the remarkable applications of KTaO$_3$ in the field of spintronics and prospects of studying emergent physics close to the ferroelectric quantum critical point, several successful attempts have been made in recent times to introduce free carriers in the bulk as well as at the surface or interface of KTaO$_3$ and a wide range of phenomena ranging from topological Hall effect to 2D superconductivity have been reported[10–13]. However, so far there has been no report about glassy dynamics in electron-doped KTaO$_3$. For the current investigation, metallic samples have been prepared by introducing oxygen vacancies in pristine single crystalline (001) oriented KTaO$_3$ substrate (for details see "Methods" section and references[14,15]). These samples were found to exhibit quantum oscillations below 10 K[14], which is signature of a good metal with well-defined Fermi surface. To further testify this, we have also computed temperature-dependent mean free path of electrons ($l_e$) within Drude-Boltzmann picture. Figure 1d shows the corresponding plots for degenerate and non-degenerate case. A dotted vertical line marks the temperature above which $l_e$ becomes shorter than the inverse of Fermi wave-vector ($k_F$) and the sample crosses over from good metal to a bad metal phase[16]. In the current work, observation of glassy dynamics is inherently constrained to temperatures which is much lower than crossover temperature to bad metal phase and hence for all practical purposes our electron doped KTaO$_3$ system can be considered as a good metal with well-defined scattering.

### Demonstration of glassy dynamics

Oxygen vacancy creation in KTaO$_3$ not only adds free electrons but also leads to the formation of highly localized defect states. To determine the exact position of the defect states, valence band

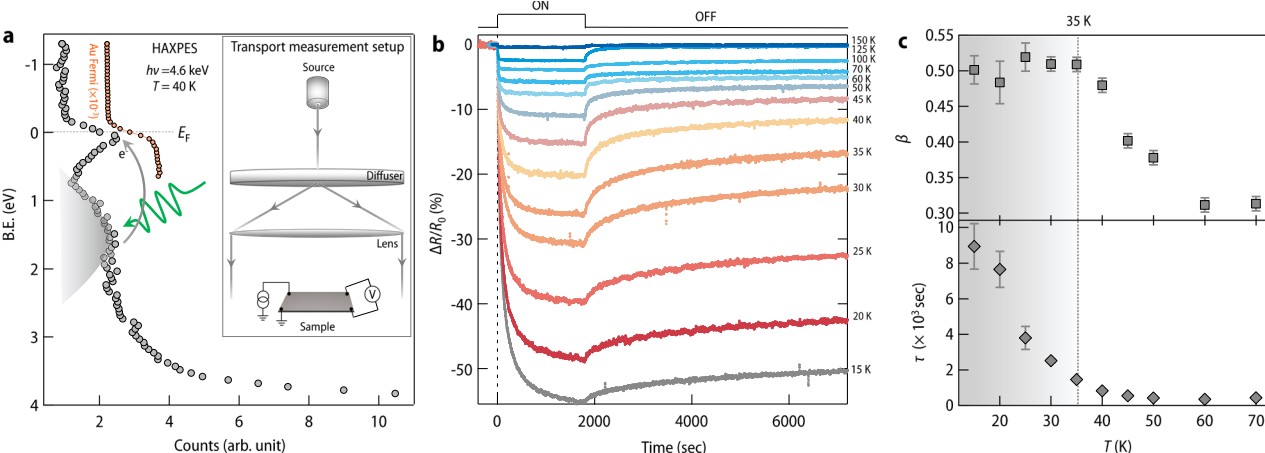

**Fig. 2 | HAXPES and transport measurements under light. a** Near Fermi level electronic states measured at 40 K using hard X-ray photoelectron spectroscopy on the KTaO$_{3-\delta}$ sample. Binding energy was corrected by taking Au Fermi level as a reference. In the figure, Au Fermi level intensity has been reduced by thousand times and shifted rightward for comparison. Inset shows the setup for transport measurement under light illumination (see "Methods" for more details). Green arrow shows the incoming photon which excites trapped electrons to the conduction band (shown with a slate-gray color). A slate-gray shade has been used to highlight the presence of defect states. **b** Temporal evolution of resistance under green light illumination ($\lambda$ = 527 nm, power = 145 µwatt) for 30 min measured at several fixed temperatures. After 30 min, resistance relaxation was observed in dark for the next 1.5 h. For comparative analysis, change in resistance has been

converted into relative percentage change ($\Delta R/R_0$) × 100. Additional measurements with red light have been shown in Supplementary Note 5 and Supplementary Fig. 5. **c** Temperature dependence of stretching exponent ($\beta$) and relaxation time ($\tau$) obtained from fitting of resistance relaxation in light off stage with a stretched exponential function. A slate-gray shade has been used to highlight the distinct behavior of $\beta$ and $\tau$ below 35 K. The error bar in $\beta$ and $\tau$ has been estimated from the variation of corresponding parameters which results in similar fitting. We further emphasize that, while the value $\beta$ has been obtained by fitting the data for 1.5 h, the same value is obtained even when the data is fitted for a longer period of up to 24 h (Supplementary Note 6 and Supplementary Fig. 6). Source data are provided as a Source data file.

spectrum has been mapped out by using hard X-ray photoelectron spectroscopy (HAXPES) at P22 beamline of PETRA III, DESY (see "Methods" for more details). Apart from the well-defined quasi-particle peak, mid-gap states centered at 1.6–1.8 eV is observed (Fig. 2a), which arises due to the clustering of oxygen vacancies[15].

In this work, we utilize these defect states to perturb the system by selective excitation of trapped electrons to the conduction band via sub-bandgap light illumination. Subsequently, the system's response is studied by monitoring the temporal evolution of the electrical resistance at a fixed temperature (see inset of Fig. 2a for transport measurement set-up). For each measurement, the sample was first cooled down to the desired temperature. Once the temperature stabilizes, the system was driven out of equilibrium by shining light for half an hour, thereafter resistance relaxation was observed for the next 1.5 h in dark condition. For the next measurement, the sample was heated to room temperature where the original resistance is recovered. Figure 2b shows one set of data recorded with green light ($\lambda$ = 527 nm) at several fixed temperatures ranging from 15 K to 150 K.

At first glance, Fig. 2b reveals a striking temperature dependence of the photo-doping effect (also see Supplementary Note 2 and Supplementary Fig. 2) and the way the system relaxes after turning off the light is also found to be strongly temperature dependent. Further analysis reveals that in the off-stage, the resistance relaxes in a stretched exponential manner [exp($-(t/\tau)^\beta$) where $\tau$ is the relaxation time and $\beta$ (stretching exponent) < 1 (see Supplementary Fig. 3 for fitting of few representative data)]. Such stretched exponential relaxation is very often considered as a signature of glassy dynamics and has been observed in variety of glassy systems[4,17,18]. In glass physics, it is commonly accepted that such stretching is due to a distribution of relaxation times arising from the disorder-induced heterogeneity[19,20]. In the present case, the distribution of relaxation times would correspond to multiple relaxation channels for the electron-hole recombination[21]. The microscopic origin behind the spatial separation of electron-hole pair and resultant non-exponential relaxation will be discussed later. The temperature evolution of $\beta$ and $\tau$ obtained from

the fitting further reveals a substantial increase in relaxation time below 50 K with a power law behavior ($\tau \sim T^{-2.8}$, see Supplementary Note 4 and Supplementary Fig. 4) followed by constant $\beta \approx 0.5$ below 35 K (Fig. 2c). This observation is remarkable given the fact that this crossover temperature roughly coincides with the onset of quantum fluctuation in undoped KTaO$_3$.

One critical test to confirm glassiness is the observation of the aging phenomenon wherein the system's response depends on its age[22]. More precisely, older systems are found to relax more slowly than younger ones. To examine this, we prepared the system of desired age by tuning the duration of light illumination ($t_{ill}$), and measurements similar to that shown in Fig. 2b were performed at 15 K (Fig. 3a). For further analysis, we only focus on relaxation after turning off the light. In Fig. 3b we plot the change in resistance in the off-stage ($\Delta R_{off}$) normalized with a total drop in resistance ($\Delta R_{ill}$) at the end of illumination. Evidently, with increasing $t_{ill}$ system's response becomes more and more sluggish. More precisely, $\tau$ obtained from the fitting is found to scale linearly with $t_{ill}$ (inset of Fig. 3c). This is the defining criteria for simple or full aging[22] which is more clear in Fig. 3c where all the curves can be collapsed to a universal curve by normalizing the abscissa by $t_{ill}$. A careful look at Fig. 3c reveals that for larger values of $t_{ill}$, curves start to deviate from the universal scaling. This is much clear in Fig. 3d which contains a similar set of scaling data for another sample with a lower carrier concentration. Such an observation is consistent with the criteria for aging that $t_{ill}$ should be much less than the time required to reach the new equilibrium under perturbation. As evident from Fig. 3a and inset of Fig. 3d, above a critical value of $t_{ill}$ there is little change in resistance upon shining the light any further. This signifies that the system is closer to its new equilibrium and hence aging ceases to hold at a higher $t_{ill}$.

## Presence of polar nano regions & importance of quantum fluctuations

As mentioned earlier, the glassy behavior of electrons in conventional electron glasses results from the competition between disorder and Coulomb interactions and is only applicable in strongly localized

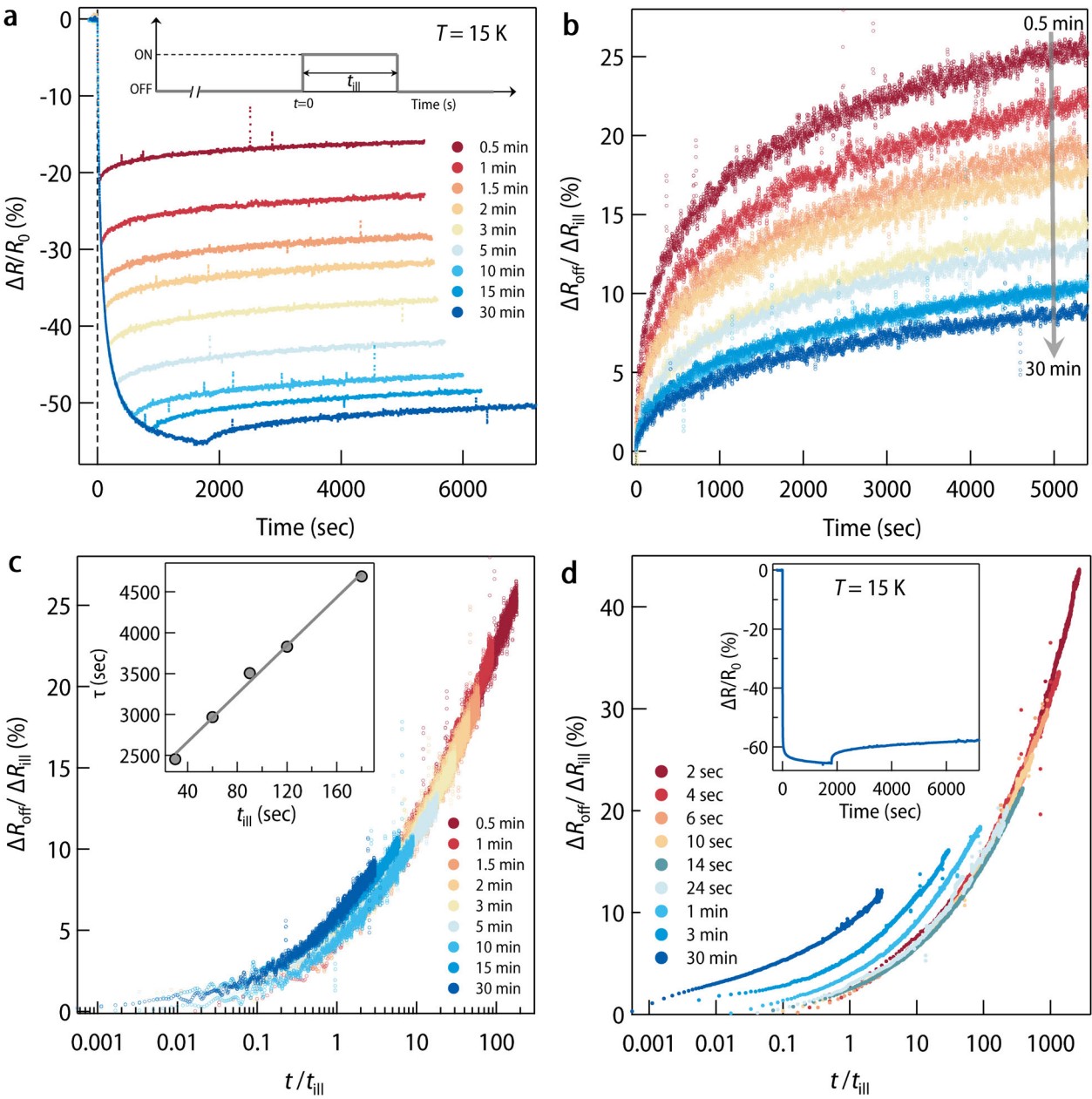

**Fig. 3 | Aging phenomena. a** Relative percentage change in resistance $(\Delta R/R_0) \times 100$ measured at 15 K for different $t_{ill}$ ranging from 0.5 to 30 min. After turning off the light, resistance relaxation was observed for the next 1.5 h in every case. **b** Temporal evolution of resistance relaxation in off-stage for different values of $t_{ill}$. Since the amount of doping depends on the duration of light illumination, the change in resistance in the off-stage ($\Delta R_{off}$) has been normalized with a total drop in resistance ($\Delta R_{ill}$) at the end of illumination[63]. **c** Upon re-scaling the time axis with $t_{ill}$, all the curves in off-stage (except $t_{ill} = 30$ min) are found to fall on a universal curve.

Inset shows the linear relationship between $\tau$ and $t_{ill}$ which is the defining criteria for simple/full aging. **d** Simple aging observed for another sample at 15 K. Inset shows one representative resistance relaxation data at 15 K. The sheet resistance of this sample is approximately 22 kΩ/sq. (at room temperature) which is much larger than the one discussed throughout this manuscript which has around 200 Ω/sq. emphasizing that the oxygen vacancy concentration for this sample is much lower. Source data are provided as a Source data file.

regime[3-5]. As glassiness is observed within a good metal regime ($k_F l_e > 1$) in our oxygen-deficient KTaO$_3$ samples, we need to find other processes responsible for the electron-hole separation and complex glassy relaxations in the present case. In the context of electron doping in another well-studied quantum paraelectric SrTiO$_3$, a large lattice relaxation (LLR) model involving deep trap levels[23] has been associated as a dominant cause for prohibiting electron-hole recombination[24,25]. However, our analysis of $\tau$ vs. $T$ (Supplementary Note 7 and Supplementary Fig. 7) does not support the applicability of the LLR model in the present case. In sharp contrast, we will conclusively demonstrate here that the effective charge separation in such systems directly

correlates with the appearance of polar nano regions (PNRs), which arise as a direct consequence of the defect dipoles present in a highly polarizable lattice of quantum paraelectric[26].

In an ordinary dielectric host, an electric dipole can polarize the lattice only in its immediate vicinity and hence the correlation length ($r_c$) is generally of the order of unit cell length which further remains independent of temperature[27]. However, the situation is drastically different in highly polarizable hosts such as KTaO$_3$ where the magnitude of $r_c$ is controlled by the polarizability of the lattice which is inversely proportional to the soft mode frequency ($\omega_s$). Since $\omega_s$ decreases with decreasing temperature, $r_c$ becomes large at lower

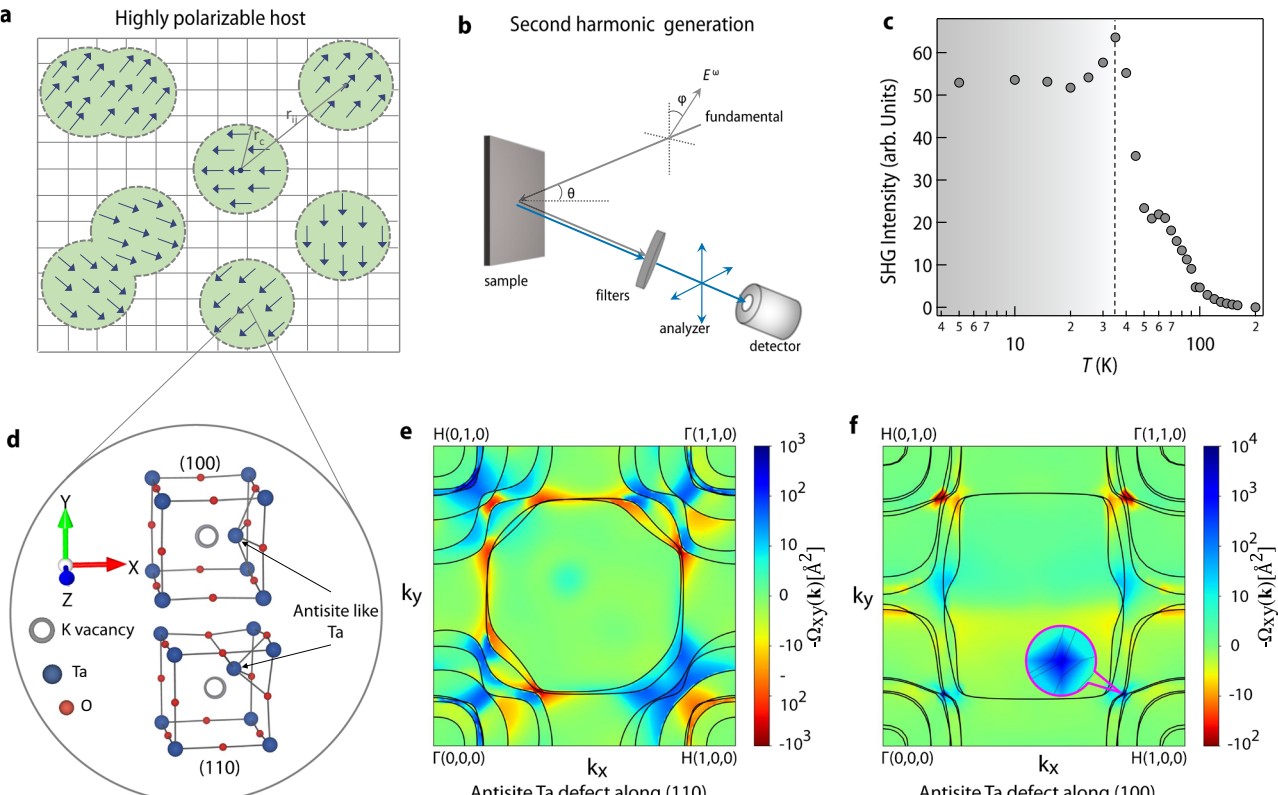

**Fig. 4 | Polar nano regions, second harmonic generation measurement and density functional theory. a** A schematic depicting real space random distribution of polar nano regions in a highly polarizable host lattice. **b** A schematic of the experimental setup for measuring optical second harmonic generation signal in reflection geometry (for more details see "Methods" section). **c** Temperature-dependent second harmonic generation intensity measured on $KTaO_{3-\delta}$ sample. A slate-gray shade has been used to highlight the distinct second harmonic generation signal below 35 K. **d** A portion of the relaxed structure of supercell of size $2 \times 2 \times 2$ of $KTaO_3$ with Ta off-centering along the (100) and (110) direction around K vacancy. For more details, see the "Methods" section, Supplementary Note 12, and

Supplementary Fig. 12. **e** Plot of Berry curvature $\Omega_{xy}(k)$ (shown in color-map) for $k_z = 0$ and bands (solid contours) intersecting the Fermi surface for the system having an antisite-like Ta defect along (110) direction. **f** Same as (**e**). for the system with an antisite-like Ta defect along (100) direction. The inset shows the spin-orbit induced avoided crossing of bands yielding large contributions to the Berry curvature. We get average polarization of 5.8 μC/cm² and 11.1 μC/cm² for Ta antisite-like defect along (110) and (100) respectively for the supercell of size $2 \times 2 \times 2$. We also note that, we do not observe any polarization in pristine $KTaO_3$. Source data are provided as a Source data file.

temperatures. As a result, PNRs spanning several unit cells are formed around the defect dipole which is randomly distributed in the lattice (Fig. 4a).

While PNRs are quite well established in the insulating regime (Supplementary Note 8 and Supplementary Fig. 8), they are expected to vanish in the metals due to screening effects from free electrons. Surprisingly, several recent experiments have reported that PNRs can exist even in the metallic regime[28–32]. Motivated by these results, we have carried out temperature-dependent optical second harmonic generation (SHG) measurement (Fig. 4b) which is a powerful technique to probe PNRs[33]. Figure 4c shows the temperature evolution of SHG intensity which is directly proportional to the volume density of PNRs. As evident, no appreciable SHG signal is observed at room temperature, however, a strong signal enhancement is observed below 150 K, signifying the appearance of PNR below 150 K in our metallic sample. Since this onset temperature exactly coincides with the temperature below which an appreciable photo-doping effect is observed in our transport measurements (Supplementary Note 9 and Supplementary Fig. 9), we believe that the internal electric field generated around such PNRs is the major cause behind driving apart the photo-generated electron-hole pairs in real space. Another notable observation is that the SHG intensity is independent of the temperature below 35 K. This immediately reminds of the regime of quantum fluctuation which enforces a constant value of $\omega_s$ below 35 K[9]. Since $r_c \propto \omega_s^{-1}$[9], our SHG measurement conclusively establishes that quantum fluctuation-

stabilized soft polar mode is retained even in metallic $KTaO_3$[34] (also see Supplementary Note 10).

We have also conducted an investigation into the primary defect dipoles responsible for the creation of PNRs in our samples. Considering that our samples were prepared through high-temperature annealing within an evacuated sealed quartz tube[15], there is a possibility of K vacancies due to its high volatility. The presence of a certain degree of K vacancy has been indeed observed in our HAXPES measurements (Supplementary Note 11 and Supplementary Fig. 11). It is widely established that, in $ABO_3$ systems, the off-centering of substitute $B$ atom ($B$ antisite-like defect) in the presence of $A$ atom vacancy leads to a macroscopic polarization even in a non-polar matrix[33,35–38]. Our density functional theory calculations (details are in the "Methods" section) considering Ta antisite-like defect has found significant Ta off-centering along [100] and [110] in the presence of K vacancy (Fig. 4d). Further, we have also computed induced polarization in the system following the modern theory of polarization where the change in macroscopic electric polarization is represented by a Berry phase[39–43] and the non-zero Berry curvature i.e., Berry phase per unit area, is taken as a signature of finite polarization in the material. In Fig. 4e, f, we have plotted the Berry curvature in the plane $k_z = 0(\Omega_{xy}(k))$ for the Ta off-centering along [110] and [100], respectively. From the figure, it is clear that the large contributions to the Berry curvature are due to the avoided crossings of bands at the Fermi surface (see inset of Fig. 4f) which are induced by spin-orbit coupling.

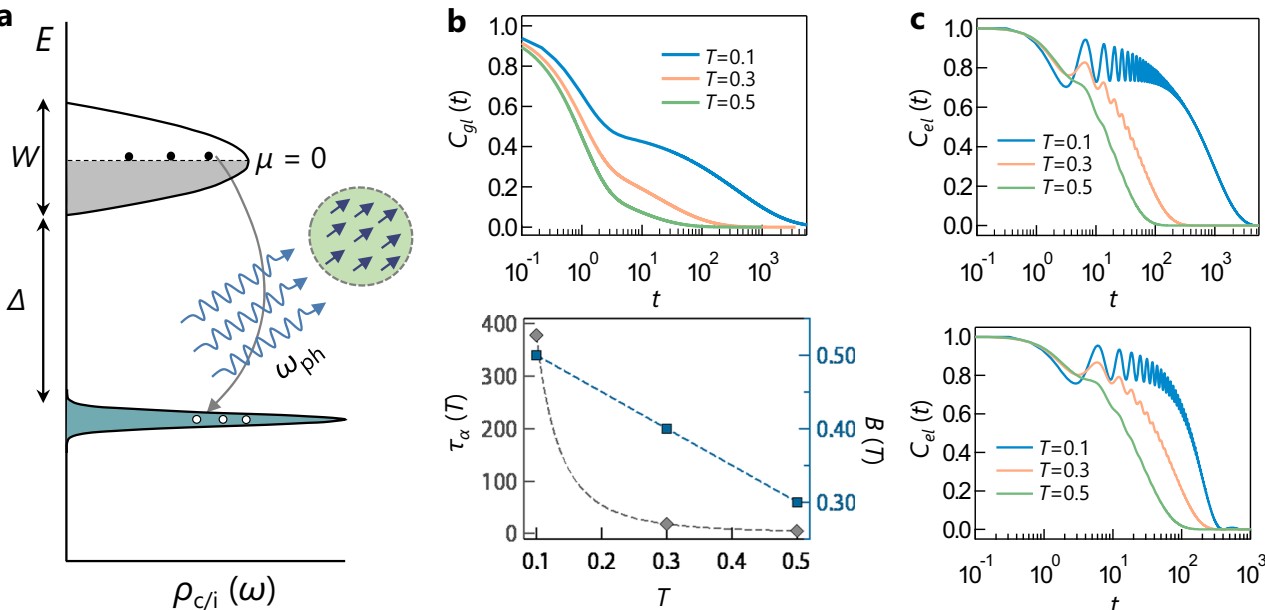

**Fig. 5 | Theoretical calculations. a** A schematic of band diagram considered for computing inter band transitions when the conduction electron is coupled to glassy background from randomly oriented polar nano regions. **b** (Upper panel) The glass correlation function $C_{gl}(t)$ vs. $t$ for three temperatures for $\tau_s(T) = 1$, $b = 0.5$. (Lower panel) Plot of $\tau_\alpha(T) \sim T^{-2.8}$ vs. $T$ (left y axis, in units of $\tau_s$) and $B(T)$ (right y axis) vs. $T$ ($A = 1 - B$) for three temperatures. **c** (upper panel) The density-density correlation function of conduction electron $C_{el}(t)$ vs. $t$ for three temperatures for flat ($W = 0$) conduction band. (lower panel) $C_{el}(t)$ vs. $t$ for three temperatures for semicircular conduction band with $W = 0.01$. Source data are provided as a Source data file.

## Phenomenological model to understand glassy dynamics

We now discuss a possible mechanism for the emergent glassy dynamics in a metal where conduction electrons coexist with PNRs. Since the glassiness in the present case is observed in the dynamical relaxation of excess injected carriers in the conduction band, it is necessary to have a thorough understanding of the relaxation processes happening against the backdrop of randomly oriented PNRs. In indirect-band semiconductors like KTaO₃ which have strong electron-lattice and defect-lattice coupling, the relaxation should be predominantly nonradiative and manifest itself as the emission of several low-energy phonons[44]. Further, since inter-band electron-phonon matrix element due to acoustic phonons are negligible, the electron-hole recombination may primarily involve soft polar modes at low temperatures, although it is not the lowest energy phonon[45]. While such a multi-phonon inter-band transition could lead to multichannel relaxation with large time scales[44], it can never give rise to collective glassy behavior.

Instead, we suggest the following scenario for the observed glassiness. As was previously mentioned, there is clear evidence that the internal electric field around PNRs has a significant impact on electron-hole recombination in our sample. It has been demonstrated previously[26] that the random interactions between PNRs (in the limit of dilute defect dipoles) leads to a dipole glass at low temperatures in KTaO₃. Recently, long-lived glass-like relaxations in SHG and Kerr signals were observed in pristine KTaO₃ at temperatures below 50 K and was attributed to dipolar correlations among PNRs, further highlighting the potential role of PNRs in influencing relaxation properties[46,47]. Electrons and holes being charged particles would immediately couple to the complex electric field from the dipole glass and hence there is a chance that the glassy background of PNRs can induce glassiness to the free carriers in the system.

In order to study such a possibility, we consider a theoretical model with the Hamiltonian,

$$H = H_{el} + H_{gl} + H_{el-gl}, \quad (1)$$

where $H_{el} = -\sum_{ij} t_{ij} c_i^\dagger c_j - \varepsilon_0 \sum_\alpha f_\alpha^\dagger f_\alpha$ ($\varepsilon_0 > 0$) describes the electronic part in terms of creation and annihilation operators $c_i^\dagger, c_i$ ($i = 1, \cdots, N_c$) and $f_\alpha^\dagger, f_\alpha$ ($\alpha = 1, \cdots, N_f$) of the $N_c$ conduction and $N_f$ impurity electronic states, respectively. For our calculations, we consider various lattices and corresponding hopping amplitudes $t_{ij}$ to describe several different energy dispersions for the conduction band, e.g., a band with semicircular DOS $g(\epsilon) = (1/2\pi)\sqrt{W^2 - \omega^2}\theta(W - |\omega|)$ with bandwidth $W$ [$\theta(x)$ is heaviside step function] (Fig. 5a), and a flat band with width $W = 0$, as discussed in the Supplementary Note 13. We take a flat impurity band at energy $-\varepsilon_0$, separated by a gap $\Delta = \varepsilon_0 - W/2$ from the conduction band minimum. We set the chemical potential $\mu = 0$, at the center of conduction band (Fig. 5a).

To model the dynamics of the glassy background, which may result from either a single PNR or randomly-distributed coupled assembly of PNRs, we consider a system with $N_g$ degrees of freedom $\{x_\mu\}$ (Supplementary Note 13). These position-like variables, related to the electric dipoles inside the PNRs, can be thought of as a multi-coordinate generalization of the usual single configuration coordinate[23,25,44] for a defect or impurity in semiconductors. Such single coordinate defect model, though can give rise to very slow relaxation of photoresistivity[23,25,44], is unlikely to lead to collective aging phenomena seen in our experiment. For our model, we thus assume a collective glassy background that gives rise to a two-step relaxation, $C_{gl}(t) = \langle x_\mu(t) x_\mu(0) \rangle = A \exp(-t/\tau_s) + B \exp[-(t/\tau_\alpha)^\beta]$ as a function of time $t$, for the dynamical correlation of $x_\mu$ at temperature $T$. The relaxation consists of a short-time exponential decay with the time scale $\tau_s$ and a long-time stretched exponential $\alpha$-relaxation with time scale $\tau_\alpha(T)$ and stretching exponent $\beta$[48,49]. We assume that $\tau_s$ is temperature independent, whereas $\tau_\alpha$ increases with decreasing temperature. We also vary the coefficients $A(T)$ and $B(T)$ with $T$ such that the relative strength of the stretched exponential part increases at lower temperatures (Supplementary Note 13). In the above form of $C_{gl}(t)$, we neglect the $\beta$ relaxation[49] which leads to a power-law decay of $C_{gl}(t)$ in the plateau region ($\tau_s \ll t \ll \tau_\alpha$), after the microscopic relaxation and before the onset of the $\alpha$-relaxation. The $\beta$ relaxation is only expected to modify finer details of electronic relaxation in Fig. 5.

The glassy PNRs lead to a transition between the conduction band and impurity state via a coupling $H_{el-gl} = \sum_{i\alpha\mu}(V_{i\alpha\mu}c_i^\dagger f_\alpha + \text{h.c})x_\mu$, where we take $V_{i\alpha\mu}$ as Gaussian complex random numbers to keep the model solvable. The coupling can arise either due to direct coupling of the electric field of the PNR to the electrons, or indirectly via coupling between PNR and electrons mediated by phonons, e.g., the soft polar optical phonon mode. To gain an analytical understanding of the electron-hole recombination dynamics we consider a limit where there are no backactions of the conduction electrons on the impurity electrons and the glass (Supplementary Note 13).

The description of the relaxation of the photo-excited electrons requires the consideration of the out-of-equilibrium quantum dynamics, which is beyond the scope of this paper. Instead, we consider an equilibrium dynamical correlation, namely the (connected) density-density correlation function $C_{el}(t) = \langle n_i(t)n_i(0)\rangle - \langle n_i(0)\rangle^2$ as a function of time $t$ for the density $n_i = c_i^\dagger c_i$ of the conduction electron. In our experiment, the resistivity decreases due to the increase of the density of conduction electrons via photo-excitations and relaxes through the relaxation of these excess carriers to the impurity states. The correlation function $C_{el}(t)$ characterizes a similar relaxation process, albeit close to the thermal equilibrium. The correlation function $C_{el}(t)$ can be computed exactly at a temperature $T$ in the toy model (Eq. (1)). In the limit of no backaction of the conduction electrons, the electronic correlation function can be written as a convolution of the spectral function of the glass (Supplementary Note 13). As a result, the glass spectral function, which contains the information of the multiple time-scales and their non-trivial temperature dependence, may directly induce glassiness in the electronic relaxation.

To verify the above scenario, we numerically compute $C_{el}(t)$ for two-step glass correlation functions $C_{gl}(t)$, shown in Fig. 5b (upper panel) for three temperatures $T = 0.5, 0.3, 0.1 \ll \Delta$, where the temperature dependence is parameterized by $B(T)$ $(A = 1 - B)$ and $\tau_\alpha(T)$ (Fig. 5b (lower panel)). We take a temperature independent exponent $\beta = 0.5$ and $\tau_\alpha(T) \sim T^{-2.8}$, consistent with our experimental results (Fig. 2c). Here $\tau_s^{-1}$ $(\tau_s)$ is set as the unit of energy (time) $(\hbar = 1)$. The calculated $C_{el}(t)$ is plotted in Fig. 5c for a flat conduction band $(W = 0)$ (upper panel) and a semicircular conduction band DOS with $W = 0.01$, and electron-glass coupling $V = 0.3$. As shown in Fig. 5c, we find that for a local glassy bath, whose bandwidth is comparable or larger than the electronic energy scales $W$ and $\Delta$, the complex glassy correlation, namely the two-step relaxation with long and non-trivial temperature-dependent time scale, is also manifested in the electronic relaxation.

## Outlook

While we do observe complex relaxations in such a toy model, these are still weak in contrast to the actual experimental results since our calculations only capture glassy two-step electronic relaxation via equilibrium dynamical correlations, whereas the actual experimental electron dynamics take place in a strongly non-equilibrium condition. We expect that a more realistic non-equilibrium theory considering the direct coupling with a real space distribution of PNRs embedded in the sea of conduction electrons would yield a strong glass. This would further demand intricate knowledge about the nature of interactions in the presence of PNRs which is still a subject of debate. Interestingly, these questions form the basis for understanding the nature of conduction in an interesting class of materials known as polar metals[30,31,50], and hence we believe that our finding of glassy relaxations in presence of PNRs will be crucial in building the theory of conduction in quantum critical polar metals. Further, the observation of glassy dynamics deep inside the good metallic regime is in sharp contrast with the conventional semiconductors where glassy relaxation ceases to exist just before the insulator-metal transition (IMT) is approached from the insulating side. This raises the question about the envisaged role of glassy freezing of electrons as a precursor to IMT apart from the Anderson and Mott localization[3,51].

## Methods

### Sample preparation

Oxygen-deficient $KTaO_3$ single crystals were prepared by heating as received (001) oriented pristine $KTaO_3$ substrate (from Princeton Scientific Corp.) in a vacuum-sealed quartz tube in presence of titanium wire. For more details we refer to our previous work[14,15].

### Dielectric measurement

Temperature-dependent dielectric measurement was performed in a close cycle cryostat using an impedance analyzer from Keysight Technology Instruments (Model No. E49908).

### Transport measurement

All the transport measurements were carried out in an ARS close cycle cryostat in van der Pauw geometry using a dc delta mode with a Keithley 6221 current source and a Keithley 2182A nanovoltmeter and also using standard low-frequency lock-in technique. Ohmic contacts were realized by ultrasonically bonding aluminum wire or by attaching gold wire with silver paint.

### Light set-up

Light of the desired wavelength was passed through the optical window of the close cycle cryostat from ARS. A home-built setup consisting of a diffuser and lens was used to make light fall homogeneously over the sample (see inset of Fig. 2a). Commercially available light-emitting diodes from Thor Labs were used as a light source. Incident power on the sample was measured with a laser check handheld power meter from coherent (Model No: 54-018).

### HAXPES measurement

Near Fermi level and K 2p core level spectra were collected at Hard X-ray Photoelectron Spectroscopy (HAXPES) beamline (P22) of PETRA III, DESY, Hamburg, Germany using a high-resolution Phoibos electron analyzer[52]. Au Fermi level and Au 4f core level spectra collected on a gold foil (mounted on the same sample holder) were used as a reference for making the correction to the measured kinetic energy. The chamber pressure during the measurement was $\sim10^{-10}$ Torr. An open cycle Helium flow cryostat was used to control the sample temperature.

### Second harmonic generation measurement

SHG measurements were performed under reflection off the sample at a 45-degree incidence angle. A $p$-polarized 800 nm beam from a Spectra-Physics Spirit-NOPA laser was used as the fundamental beam (pulse width: 300 fs, repetition rate: 1 MHz), and was focused on the sample surface. The $p$-polarized SHG intensity generated by the sample, was measured by a photo multiplier tube. The sample temperature was controlled by a helium-cooled Janis 300 cryostat installed with a heating element.

### Density functional theory

The noncolinear density functional theory calculations were carried out using the QUANTUM ESPRESSO package[53]. In this calculation optimized norm-conserving pseudopotentials[54–56] were used and for the exchange-correlation functional[57] we have incorporated Perdew, Burke, and Ernzerhof generalized gradient approximation (PBE-GGA). For the unit cell, the Brillouin zone was sampled with $8 \times 8 \times 8$ $k$-points. The wave functions were expanded in plane waves with an energy up to 90 Ry. Since the effect of SOC in $KTaO_3$ is quite remarkable, we have employed full-relativistic pseudopotential for the Ta atom. The structural relaxations were performed until the force on each atom was reduced to 0.07 eV/Å. The Berry phase calculations are carried out as implemented in the Wannier90 code[58,59]. We have used a $41 \times 41$ 2D $k$-mesh for the Berry curvature calculations. No significant change in the result is observed on

increasing the k-mesh up to 101 × 101 (see Supplementary Note 8 for further details).

## Data availability

The authors declare that the data supporting the findings of this study are available within the main text and its Supplementary Information and at https://doi.org/10.6084/m9.figshare.25549483 Other relevant data are available from the corresponding author upon request.

## Code availability

The code that support the findings of this study are available from the corresponding author upon request.

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

## Acknowledgements

The work is funded by SERB, India, by a core research grant CRG/2022/001906 to S.M. SM is also funded by Quantum Research Park, which is a project administered by FSID, IISc with support from KITS, Government of Karnataka. The authors acknowledge the uses of central facilities of Department of Physics, IISc supported by DST-FIST program. Portions of this research were carried out at the light source PETRA III DESY, a member of the Helmholtz Association (HGF). We would like to thank Dr. Anuradha Bhogra and Dr. Thiago Peixoto for their assistance at beamline P22. Financial support by the Department of Science & Technology (Government of India) provided within the framework of the India@DESY collaboration is gratefully acknowledged. S.H. and V.G. acknowledge support from the US Department of Energy under grant no. DE-SC0012375 for temperature-dependent second-harmonic generation measurements. SB acknowledges support from SERB (CRG/2022/001062), DST, India. S.K.G. and M.J. gratefully acknowledge Supercomputer Education and Research Centre, IISc for providing computational facilities SAHASRAT and PARAM-PRAVEGA. S.K.G. acknowledges DST-Inspire fellowship (IF170557). S.K.O. acknowledges the wire bonding facility at the Department of Physics, IISc Bangalore and thanks Shivam Nigam for the experimental assistance. S.K.O. and S.M. thank Professor D. D. Sarma for giving them access to the quartz tube sealing and dielectric measurement setup in his lab.

## Author contributions

S.M. conceived and supervised the project. S.K.O., S.M., and S.K. came up with all experimental plans to confirm the glassy behavior. S.K.O., S.H., and J.M. carried out transport measurements. PM helped in making the photoconductivity setup. S.H. performed SHG measurements under the supervision of V.G. S.K.O., P.M., A.G., and C.S. carried out HAXPES measurements. S.K.O. performed all the analysis of transport measurements and HAXPES. S.K.G. and M.J. performed DFT calculations. S. Bera, S.K., and S.B. provided phenomenological model calculations. S.K.O., S. Bera, S.B., and S.M. wrote the manuscript with inputs from other authors. All authors discussed the results and participated in finalizing the manuscript.

## Competing interests

The authors declare no competing interests.
