## [Peer Review File · Nature Communications]

REVIEWER COMMENTS

Reviewer #1 (Remarks to the Author):

I carefully read the authors' response to my criticisms expressed in the first report. They provided very careful explanations and responses to all my questions. They provide a detailed response to my query about extraction 3D Drude model parameters from their transport measurements. Most importantly, the authors explain the importance of their new results compared to the more established phenomenology of electron glassiness in disordered insulating systems. My original reading of their work was as something that could be placed and interpreted alongside the established phenomenology. I now understand that their findings should be contrasted with the electron glassiness in insulating insulators as new electron glassiness in a conducting case. The authors also reduced the breadth of their original claim of electron glassiness in a "good metal" and replaced it with a claim of glassiness "in the good metal regime of electron-doped KTaO_3 ", which removes a large part of my original objections. Finally, they also convincingly answer my question about deducing the presence of the soft mode from the SHG measurement and provide new Raman data in the Supplemental material to confirm the presence of the soft mode in doped KTO. In summary, I believe that the authors have discovered a new regime of glassy electron photoconductive response in the conducting electron-doped material KTO. I believe that the manuscript is technically sound and that the potential importance and impact of findings merit their publication in a high visibility journal like Nature Communications.

Reviewer #2 (Remarks to the Author):

I have read the revised version of the manuscript, transferred to Nature Communications, and still think that it has various weak points and does not represent an important advance in glass physics. As discussed in my previous report, time-dependent resistivity experiments were already done in other electron glasses (ref. 20). The investigated system is very special: a known quantum paraelectric with oxygen vacancies to make it metallic and with additional potassium vacancies, assumed to generate polar nanoregions. This is of no relevance to other glass-forming systems or to the glass transition in general. The presented model involves many assumptions (as admitted by the

authors in their response letter). Finally, there are still various problems concerning the interpretation and analysis of the results as detailed below. Thus, I do not recommend publication of the manuscript in Nature Communications.

1. In the revised manuscript, the authors have added the passage:

"Furthermore, due to the light mass of electrons, electron glasses are highly susceptible to quantum fluctuations. This aspect introduces additional complexities in understanding the behavior of electron glasses [8], a facet that is often overlooked in the context of conventional glass formers [9]."

As I already have remarked in my previous report, the quantum fluctuation occurring in electron glasses are not of any relevance for "conventional glass formers". None of the currently considered models of the glass transition like the Adam-Gibbs model (and its modern extensions) or the mode-coupling theory have to invoke quantum fluctuations to describe the experimental data.

2. In my original report, I have suggested checking the quality of the "pristine" sample by comparing the absolute values of ϵ' (Fig. 1b) to literature values. In the revised manuscript the authors now show ϵ' at lower frequency, leading to somewhat higher absolute values. They state in the figure caption: "We further note that the value of ϵ' for our sample appears to be slightly lower than the reported values [10]." The limiting low-temperature value of ϵ' in [10] is about 4600, while the authors' revised value is about 2700. This is not "slightly lower". The saturation of ϵ' at low temperatures, $<10\text{K}$, also is less pronounced than in literature data. These discrepancies seem to indicate problems with the sample quality.

3. In response to my criticism concerning the discussion of the temperature-dependent relaxation time, in Supplementary Figure 4 the authors now show fits of these data by a power law, τ proportional to T^{-a} . This is very unusual for a relaxation process. In the Supplementary Section 4, the authors state: "We also emphasize that such power-law divergence of relaxation time has been also discussed theoretically in context of α relaxation in glasses" and refer to lecture notes from a summer school in 2002 (ref. 9 in the Supplementary). However, searching for "power law" in that work reveals that there the critical power law predicted by the mode-coupling theory (MCT) is discussed, i.e. τ proportional to $(T-T_c)^{-\gamma}$ [eq. 3.22 in that work]. In contrast to the authors' power law, it predicts a divergence at T_c , a kind of idealized glass-transition temperature above T_g assumed within this theory. It is known to roughly describe the α relaxation of glass formers at high temperatures, above the critical temperature T_c of MCT, i.e., deep in the liquid regime where relaxation times are very short. There is clearly no relation to the power law claimed by the authors for their relaxation times of order 300-10000s. Thus, using the latter to describe the data is unjustified. Any calculations based on this power law (Fig. 5) are therefore very questionable.

4. Concerning the calculation of the electrons' mean free path shown in Fig. 1d, the authors state in their response letter "that a major portion of the calculation was already presented in our earlier publication". Then they should cite this publication in the manuscript, in the paragraph where the mean free path is discussed.

5. In my first report, I have criticized the quite arbitrary assumption of a two-step relaxation involving a fast exponential and a slower stretched decay in their model. In their response and the manuscript, the authors invoke an article by Walter Kob on arXiv (ref. 53) discussing the mode-coupling theory (MCT) of the glass transition. In the revised manuscript, the authors now associate the exponential part in their assumed two-step relaxation with the microscopic process considered in the MCT. However, this process is very fast, typically of the order of phonon frequencies, and certainly not relevant for the slow time-dependent effects detected in the authors' experiments. It is not clear what the authors want to prove with their model. They put in a two-step relaxation, and they get out a two-step relaxation. Their data reveal a one-step relaxation.

Reviewer #3 (Remarks to the Author):

While I still have some reservations regarding the use of "good metal" to describe doped KTaO_3 , I can accept the authors' reasoning and do not see any further barriers to publication of this article in Nature Communications. I do believe the authors have sufficiently narrowed, clarified, and justified their claims in response to the reports by all referees.

Reviewer #4 (Remarks to the Author):

I am satisfied that the authors have addressed my comments thoroughly and have no hesitation in recommending the manuscript for publication. I want to express my gratitude to the authors for their good grace and applaud them for an excellent piece of work.

NCOMMS-24-00310-T: Quantum fluctuations lead to glassy electron dynamics
in the good metal regime of electron doped KTaO_3

Reply to Reviewer 1

Reviewer: *I carefully read the authors' response to my criticisms expressed in the first report. They provided very careful explanations and responses to all my questions. They provide a detailed response to my query about extraction 3D Drude model parameters from their transport measurements. Most importantly, the authors explain the importance of their new results compared to the more established phenomenology of electron glassiness in disordered insulating systems. My original reading of their work was as something that could be placed and interpreted alongside the established phenomenology. I now understand that their findings should be contrasted with the electron glassiness in insulating insulators as new electron glassiness in a conducting case. The authors also reduced the breadth of their original claim of electron glassiness in a "good metal" and replaced it with a claim of glassiness "in the good metal regime of electron-doped KTaO_3 ", which removes a large part of my original objections. Finally, they also convincingly answer my question about deducing the presence of the soft mode from the SHG measurement and provide new Raman data in the Supplemental material to confirm the presence of the soft mode in doped KTO. In summary, I believe that the authors have discovered a new regime of glassy electron photoconductive response in the conducting electron-doped material KTO. I believe that the manuscript is technically sound and that the potential importance and impact of findings merit their publication in a high visibility journal like Nature Communications.*

Reply: We are delighted to find that the reviewer now completely agrees with our claim of glassy electron dynamics in the good metal regime of electron-doped KTaO_3 and is fully satisfied with our responses to all of his/her questions and comments. We are also very happy to receive recognition and appreciation for our work, as indicated by the statement, "*I believe that the authors have discovered a new regime of glassy electron photoconductive response in the conducting electron-doped material KTO. I believe that the manuscript is technically sound and that the potential importance and impact of findings merit their publication in a high visibility journal like Nature Communications.*" We thank the reviewer for reviewing our paper and recommending our work for publication in Nature Communications.

Reply to Reviewer 2

Reviewer: *I have read the revised version of the manuscript, transferred to Nature Communications, and still think that it has various weak points and does not represent an important advance in glass physics. As discussed in my previous report, time-dependent resistivity experiments were already done in other electron glasses (ref. 20). The investigated system is very special: a known quantum paraelectric with oxygen vacancies to make it metallic and with additional potassium vacancies, assumed to generate polar nanoregions. This is of no relevance to other glass-forming systems or to the glass transition in general. The presented model involves many assumptions (as admitted by the authors in their response letter). Finally, there are still various problems concerning the interpretation and analysis of the results as detailed below. Thus, I do not recommend publication of the manuscript in Nature Communications.*

Reply: First and foremost, we thank the reviewer for thoroughly examining our revised manuscript and our detailed responses to his/her questions and comments raised during the first round of review. In response to the reviewer’s statement that “*time-dependent resistivity experiments were already done in other electron glasses (ref. 20)*”, we would like to re-emphasize that all those previous measurements/observations were only on the insulating samples. In sharp contrast, in the present work, we provide the first evidence of glassy electron dynamics in a good metal regime which has not been reported to date. We further note that this significant discovery in the field of electron glasses has also been appreciated by the first reviewer, stating that “*I now understand that their findings should be contrasted with the electron glassiness in insulating insulators as new electron glassiness in a conducting case. In summary, I believe that the authors have discovered a new regime of glassy electron photoconductive response in the conducting electron-doped material KTO*”. This observation of glassy dynamics within a good metallic regime completely goes beyond the conventional electron glass theories, which are limited only to strongly localized regimes. This is why we had to develop a new theoretical model to explain our non-trivial results.

Regarding the reviewer’s statement that our work has no relevance to other glass-forming systems or towards the glass transition in general, we would like to reiterate that our manuscript is exclusively focused on electron glasses only. In our manuscript, we have not made any claims that our proposed mechanism is going to be relevant to other glass formers apart from electron glasses. It is not clear to us why we need to appeal to relevance of our work to conventional classical glasses or “glass transition in general” to establish the novelty of our work. In fact, the main significance of our work lies in our observation of unusual glassy phenomena, which are very different from conventional electron glasses. To this end, we would like to reemphasize that the field of electron glass is an extremely broad subject. We refer to two books on this topic: Dobrosavljevic, V. et al., *Conductor insulator quantum phase transitions* (Oxford University Press, 2012); Pollak, M. et al., *The electron glass* (Cambridge University Press, 2013) (references 3 and 4 of the main text). We have already mentioned in this reply letter that our work presents a completely new regime of electron glass.

In the following, we have responded in detail to the reviewer’s questions and have modified the manuscript accordingly. We hope that the reviewer will recommend the updated version for publication in Nature Communications.

Reviewer: *In the revised manuscript, the authors have added the passage: "Furthermore, due to the light mass of electrons, electron glasses are highly susceptible to quantum fluctuations. This aspect introduces additional complexities in understanding the behavior of electron glasses [8], a facet that is often overlooked in the context of conventional glass formers [9]." As I already have remarked in my previous report, the quantum fluctuation occurring in electron glasses are not of any relevance for "conventional glass formers". None of the currently considered models of the glass transition like the Adam-Gibbs model (and its modern extensions) or the mode-coupling theory have to invoke quantum fluctuations to describe the experimental data.*

Reply: As we have remarked in response to the referee's earlier comment, electron glasses, not the conventional glass formers, provide more relevant perspective to our present work. Thus, references to electron glasses are more appropriate in our case, as the above mentioned passage tries to convey. We agree with the referee that theories of conventional glasses typically do not need to invoke quantum fluctuations in any essential way. However, we respectfully disagree with the logic that just because quantum fluctuations are unimportant in theories of conventional glasses, such fluctuations are irrelevant in glassy phenomena in quantum materials. On the contrary, the importance of quantum fluctuations for glassy phenomena in quantum magnets and other systems have been emphasized by many previous works, for examples, Cugliandolo, L. F., and Lozano, G., *Phys. Rev. Lett.* 80, 4979 (1998); Cugliandolo, L. F., et al., *Phys. Rev. Lett.* 85, 2589 (2000); Cugliandolo, L. F., et al., *Phys. Rev. B* 64, 014403 (2001); Cugliandolo, L. F., et al., *Phys. Rev. B* 66, 014444 (2002). At the very least, quantum fluctuations can substantially reduce the glass transition temperature, some cases even to $T = 0$ K, and thus enhance quantum effects above the glass transition, in the disordered *supercooled* phase which exhibits slow and complex dynamics, like two-step relaxation. This reduction of glass transition temperature is most likely responsible for the observed unusual power-law in temperature relaxation time, $\tau_\alpha(T) \sim T^{-2.8}$ in our work, as we discuss below in response to another comment by the referee. Nonetheless, following the reviewer's concern, we have now removed the sentence "*a facet that is often overlooked in the context of conventional glass formers [9].*" from the main text. This modification does not compromise the broadness of our results in any way.

Changes made:

1. The sentence "*Furthermore, due to the light mass of electrons, electron glasses are highly susceptible to quantum fluctuations. This aspect introduces additional complexities in understanding the behavior of electron glasses [8], a facet that is often overlooked in the context of conventional glass formers [9].*" on page 1 of the main text has been modified to "*Furthermore, due to the light mass of electrons, electron glasses are highly susceptible to quantum fluctuations. This aspect introduces additional complexities in understanding the behavior of electron glasses [8].*"

Reviewer: *In my original report, I have suggested checking the quality of the "pristine" sample by comparing the absolute values of epsilon' (Fig. 1b) to literature values. In the revised*

manuscript the authors now show ϵ' at lower frequency, leading to somewhat higher absolute values. They state in the figure caption: "We further note that the value of ϵ' for our sample appears to be slightly lower than the reported values [10]." The limiting low-temperature value of ϵ' in [10] is about 4600, while the authors' revised value is about 2700. This is not "slightly lower". The saturation of ϵ' at low temperatures, $< 10\text{K}$, also is less pronounced than in literature data. These discrepancies seem to indicate problems with the sample quality.

Reply: The reviewer's concern pertains to the differences observed in the dielectric constant at low temperatures for our (001) oriented KTaO_3 substrate compared to the value reported in the paper Nature Physics 10, 367 (2014). This difference can be attributed to the inherent nature of quantum paraelectrics, which are extremely sensitive to even minute concentrations of unavoidable impurities/defect dipoles introduced into the lattice during the growth of the crystal (J. Phys.: Condens. Matter 6 4077 (1994), Phy. Rev. B 61, 6 (2000), Journal of Physics: Condensed Matter 15, R367 (2003)). To emphasize this, we had mentioned in the previous version of the manuscript and also in the referee response to the first round of referee's comments that "*We attribute this difference to the difference in the sample preparation process, which may add slight oxygen vacancies, even in the pristine, as-received crystals.*"

To illustrate this, we show in Fig. 1 [page 5 of this reply letter] the low-temperature dielectric constant of several undoped KTaO_3 samples (taken from the reference J. Phys.: Condens. Matter 6 4077 (1994)) prepared by different growth methods or even samples from the same batch with same growth methods. As evident there is drastic variation in the value of dielectric constant at low temperatures. However, despite these variations, all of these samples are identical in terms of their behavior. Below we discuss this aspect in point by point-by-point manner in light of our own data on KTaO_3 substrate.

1. To demonstrate the identical behavior of dielectric data consistent with the earlier literature, in Fig. 2 [page-6 of this reply letter], we plot comparison of the normalized temperature-dependent dielectric constant of our as received (001) oriented crystal of KTaO_3 along with one taken from the paper Journal of the Physical Society of Japan 85, 074703 (2016) where the low-temperature saturation value is close to 4400 (Note: We did not utilize the data from the reference mentioned by the reviewer (Nature Physics 10, 367 (2014)) as the data in that paper was not available at higher temperatures. Additionally, we normalized our data to 1 at 11.75 K, as this was the lowest achievable temperature in our cryostat). As evident, despite variation in low-temperature dielectric constant, these two normalized curves look identical, signifying that the behavior of our pristine KTaO_3 sample is essentially consistent with what has been discussed in the literature.
2. One of the key characteristics of quantum paraelectrics is their manifestation of a T^2 dependence of the inverse of the dielectric constant at low temperatures (Nature Physics 10, 367 (2014)). This observation is non-classical and can be only understood in terms of quantum critical theory when extended to include the effects of long-range dipolar interactions and the coupling of the electric polarization field with acoustic phonons. In Fig. 3a and 3b [page-7 of this reply letter] we show the data for pristine SrTiO_3 and

Figure 1: Low-temperature dielectric constant of several undoped KTaO_3 samples recorded at an AC frequency of 1 kHz. This image has been taken from the reference *J. Phys.: Condens. Matter* 6 4077 (1994). Here different curves correspond to different samples either prepared by different growth methods or even samples from the same batch. For more details, we refer to the reference *J. Phys.: Condens. Matter* 6 4077 (1994).

KTaO_3 , respectively taken from the reference *Nature Physics* 10, 367 (2014). Furthermore, it has been reported that this characteristic behavior remains robust even with the introduction of disorder into the system (see supplemental of *Nature Physics* 10, 367 (2014)). For instance, in Fig. 3c, we show data for a sintered SrTiO_3 sample containing a significant degree of quenched disorder. Surprisingly, despite the low-temperature dielectric constant of this SrTiO_3 sample being only 3000 [which is at least one order of magnitude smaller than that of pristine SrTiO_3 samples with dielectric constants greater than 30000 (*Journal of the Physical Society of Japan* 85, 074703 (2016))] the T^2 dependence is well preserved. In light of these findings, a similar graph is plotted in Fig. 3d for our pristine KTaO_3 sample, revealing that our samples also exhibit a similar T^2 dependence.

3. The frequency-dependent dielectric measurement on our pristine KTaO_3 reveals that the dielectric loss is an activated process (supplementary section 8 of our manuscript) similar to what has been observed in earlier literature (see *Journal of Physics: Condensed Matter* 15, R367 (2003)).
4. It has been discussed previously that, rather than focusing solely on the absolute value of the dielectric constant to assess the sample quality, attention should be paid to the dielectric loss around 40 K (*J. Phys.: Condens. Matter* 6 4077 (1994), *Journal of Physics: Condensed Matter* 15, R367 (2003)). Moreover, it has been proposed that a lower value

Figure 2: Comparison of the normalized temperature dependent dielectric constant of as received (001) oriented crystal of KTaO_3 (presented in this work) with the literature (Journal of the Physical Society of Japan 85, 074703 (2016)).

of the loss tangent indicates a better-quality sample. To facilitate comparison, in Fig. 4a [page-8 of this reply letter], we plot the loss tangent for several samples taken from the reference J. Phys.: Condens. Matter 6 4077 (1994), and in Fig. 4b, we present our data recorded at the same AC frequency. It is evident that our substrate exhibits a finite loss tangent value of around 0.2. While it is challenging to pinpoint all the factors responsible for the appearance of loss, it has been observed previously (J. Phys.: Condens. Matter 6 4077 (1994)) that the presence of oxygen vacancies significantly impacts the height of the loss tangent value. We believe that our as received pristine substrate also contains a certain degree of oxygen vacancies, which leads to a reduction in the absolute value of the dielectric constant.

Regarding the reviewer’s concern about the less pronounced saturation of epsilon at low temperatures, specifically below 10 K, we would like to clarify that our measurement only extends up to 11.75 K, which is the lowest achievable temperature in our cryostat. As a result, the saturation may appear less pronounced. With these clarifications, we hope that the reviewer would be convinced that the behavior of our pristine KTaO_3 sample is essentially very similar to what has been discussed in the literature.

We conclude our response to this question by clarifying that the focus of this manuscript is not on pristine KTaO_3 , but rather on metallic samples achieved through electron doping via oxygen vacancy creation. Furthermore, we assert the quality of our samples by highlighting the observation of Shubnikov–de Haas (SdH) oscillations in magnetoresistance (Advanced Quantum Technologies 3, 2000021 (2020)), which would not have been possible if our samples were of poor quality. Moreover, we emphasize that the effective mass extracted from the SdH analysis for our sample is identical to that reported in earlier studies on electron-doped KTaO_3 (Phys. Rev. B 19, 3041 (1979)).

Changes made:

Figure 3: Low-temperature T^2 dependence of the inverse of dielectric constant ($1/\epsilon$) for **a.** pristine SrTiO_3 (taken from the reference Nature Physics 10, 367 (2014)), **b.** pristine KTaO_3 (taken from the reference Nature Physics 10, 367 (2014)), **c.** disordered SrTiO_3 (taken from the supplemental of the reference Nature Physics 10, 367 (2014)) and **d.** as received (001) oriented crystal of KTaO_3 (presented in this work).

Figure 4: **a.** Low-temperature dielectric loss tangent ($\tan \delta$) of several undoped KTaO_3 samples recorded at an AC frequency of 1 kHz. This image has been taken from the reference *J. Phys.: Condens. Matter* 6 4077 (1994). Here different curves correspond to different samples either prepared by different growth methods or even samples from the same batch. For more details we refer to the reference *J. Phys.: Condens. Matter* 6 4077 (1994). **b.** Low-temperature dielectric loss tangent ($\tan \delta$) of as received (001) oriented crystal of KTaO_3 (presented in this work).

1. Following the reviewer’s question, the sentence “*We attribute this discrepancy to the presence of slight oxygen vacancies, even in the pristine, as-received crystal*” in the caption of Fig. 1b has been modified to “*We attribute this difference to the difference in the sample preparation process, which may add slight oxygen vacancies, even in the pristine, as-received crystals (J. Phys.: Condens. Matter 6 4077 (1994)).*”

Reviewer: *In response to my criticism concerning the discussion of the temperature-dependent relaxation time, in Supplementary Figure 4 the authors now show fits of these data by a power law, tau proportional to T^{-a} . This is very unusual for a relaxation process. In the Supplementary Section 4, the authors state: “We also emphasize that such power-law divergence of relaxation time has been also discussed theoretically in context of alpha relaxation in glasses” and refer to lecture notes from a summer school in 2002 (ref. 9 in the Supplementary). However, searching for “power law” in that work reveals that there the critical power law predicted by the mode-coupling theory (MCT) is discussed, i.e. tau proportional to $(T-T_c)^{-\gamma}$ [eq. 3.22 in that work]. In contrast to the authors’ power law, it predicts a divergence at T_c , a kind of idealized glass-transition temperature above T_g assumed within this theory. It is known to roughly describe the alpha relaxation of glass formers at high temperatures, above the critical temperature T_c of MCT, i.e., deep in the liquid regime where relaxation times are very short. There is clearly no relation to the power law claimed by the authors for their relaxation times of order 300-10000s. Thus, using the latter to describe the data is unjustified. Any calculations based on this power law (Fig. 5) are therefore very questionable.*

Reply: We more or less disagree with all the above comments by the referee, particularly the comment that the fitting of the temperature dependence of relaxation time with a power-law is unjustified. The power-law dependence over the temperature range accessed in our experiment is an experimental fact, as corroborated by the plot in Supplementary Sec. 4, Supplementary Fig. 4, and the strong deviation from Arrhenius behaviour in Supplementary Sec. 7, Supplementary Fig. 7. This is not based on any speculative theory or, for that matter, mode-coupling theory of supercooled liquid phase in conventional classical structural glasses. For our simple theoretical toy model we use this experimental input of power-law T dependence of $\tau_\alpha(T)$. We completely agree that such non-standard ($\sim T^{-2.8}$) power-law relaxation is unusual. In our opinion, this unusual observation is one of the reasons why our work is significant. The comment quoted by the referee above was to allude that the power-law temperature dependence is consistent with MCT prediction of $\tau_\alpha(T) \sim (T - T_c)^{-\gamma}$ above a dynamical crossover temperature $T_c > T_g$. Indeed, our result, $\tau_\alpha(T) \sim T^{-2.8}$, is consistent with the above MCT prediction with a very low T_c . The smallness of T_c is consistent with expected strong reduction of T_g , and, hence T_c , due to quantum fluctuations, as mentioned in response to an earlier comment by the referee. We would also like to remind the referee that relaxation time ($\sim 300 - 10000s$) here is for the electronic relaxation of photoexcited carriers, not the relaxation time of the background glass. The absolute values of electronic relaxation time depends on several non-universal factors like the smallness of the electron-glass coupling and other details of various energy scales of the material. Thus, comparison of the absolute value of the electronic relaxation time at such low temperatures with relaxation time of the supercooled liquid phase of classical glasses at much higher temperatures has no relevance for our experimental system.

Reviewer: *Concerning the calculation of the electrons' mean free path shown in Fig. 1d, the authors state in their response letter "that a major portion of the calculation was already presented in our earlier publication". Then they should cite this publication in the manuscript, in the paragraph where the mean free path is discussed.*

Reply: We thank the reviewer for this suggestion. We have now made the following change in the main text.

Change made:

1. The sentence “*For details of the calculation we refer to the Supplementary Information Section 1.*” in caption of Fig. 1d has been modified to “*For details of the calculations, we refer to Supplementary Information Section 1 which significantly overlaps with our earlier work [Advanced Quantum Technologies 3, 2000021 (2020)].*”

Reviewer: *In my first report, I have criticized the quite arbitrary assumption of a two-step relaxation involving a fast exponential and a slower stretched decay in their model. In their response and the manuscript, the authors invoke an article by Walter Kob on arXiv (ref. 53) discussing the mode-coupling theory (MCT) of the glass transition. In the revised manuscript, the authors now associate the exponential part in their assumed two-step relaxation with the microscopic process considered in the MCT. However, this process is very fast, typically of*

the order of phonon frequencies, and certainly not relevant for the slow time-dependent effects detected in the authors' experiments. It is not clear what the authors want to prove with their model. They put in a two-step relaxation, and they get out a two-step relaxation. Their data reveal a one-step relaxation.

Reply: As we have elaborately discussed in our reply to referees' comments in the earlier round of review, the two-step relaxation is, in fact, one of the most natural assumptions to describe glassy relaxation above the glass transition. This assumption is not arbitrary at all. We referred to the review article by Walter Kob (Kob, Les Houche (2002) (<https://arxiv.org/pdf/cond-mat/0212344.pdf>)) and, in particular, schematic Fig.3 in the article, not to appeal to mode-coupling theory (MCT), but to point to the most common and well-known phenomenology of glassy relaxation. Fig.3 is not based on MCT, but is a schematic summary of a huge number of computer simulation and experimental studies of glasses, where the two-step relaxation is taken as a tell-tale signature of glassy relaxation in equilibrium dynamical correlation, almost on an equal footing to aging phenomena in non-equilibrium relaxation in glasses. In theoretical works, equilibrium dynamical correlations are most often studied to probe glassy relaxation, as non-equilibrium aging phenomena are much harder to study theoretically. We have used the two-step relaxation profile, namely the signature of multiple distinct time scales, in our theoretical toy model to describe the dynamics of glassy background of polar regions, whose dynamics we cannot directly probe experimentally. Nowhere in the manuscript we have claimed that the fast microscopic relaxation plays a relevant role in the long-time relaxation of photoexcited carriers. The manifestation of two-step relaxation in equilibrium dynamical electronic density-density correlation is merely used here as an indicator or diagnostic of the glassy relaxation dynamics of excited carriers. Our experiment probes non-equilibrium aging dynamics, which much harder to study theoretically. Moreover, such theoretical exploration is beyond the scope of a predominantly experimental work. As well know in studies of glasses, unlike the two-step equilibrium relaxation, the evidence of multiple drastically different time scales in non-equilibrium dynamics comes from the aging phenomena, not from multiple steps in the relaxation profile, as in our experiment.

We indeed wish that getting the glassy behaviour in the relaxation of excited electrons was so simple as “put in a two-step relaxation”, and “get out a two-step relaxation”, as the referee put it. Even when the background bath for the electrons is glassy, the manifestation of the same glassy behaviour in electronic dynamics in a particular relaxation channel is rather complex and subtle. This can be ascertained from Eqs.(24-26), Supplementary Section 13. The *transfer* of glassy features of the background bath to the electronic relaxation depends on the interplay of the relaxation times of the glassy background with various other energy scales, like electronic electron-glass coupling, electronic bandwidth, the gap between conduction and impurity bands etc, and thus is very non-trivial. In our opinion, this is the reason why our experimental observations, that allow to probe the glassy relaxation through electronic relaxation in a particular channel, is quite unique.

Reply to Reviewer 3

Reviewer: *While I still have some reservations regarding the use of "good metal" to describe doped $KTaO_3$, I can accept the authors' reasoning and do not see any further barriers to the publication of this article in Nature Communications. I do believe the authors have sufficiently narrowed, clarified, and justified their claims in response to the reports by all referees.*

Reply: We are very happy to hear that the reviewer is fully satisfied with our response to the reports by all reviewers. We thank the reviewer for reviewing our manuscript.

Reply to Reviewer 4

Reviewer: *I am satisfied that the authors have addressed my comments thoroughly and have no hesitation in recommending the manuscript for publication. I want to express my gratitude to the authors for their good grace and applaud them for an excellent piece of work.*

Reply: We are happy to find that the reviewer is fully satisfied with our response to his/her questions and has appreciated our work by mentioning "*I want to express my gratitude to the authors for their good grace and applaud them for an excellent piece of work*". We again thank the referee for reviewing our paper.

Summary of Changes

1. The sentence "*Furthermore, due to the light mass of electrons, electron glasses are highly susceptible to quantum fluctuations. This aspect introduces additional complexities in understanding the behavior of electron glasses [8], a facet that is often overlooked in the context of conventional glass formers [9].*" on page 1 of the main text has been modified to "*Furthermore, due to the light mass of electrons, electron glasses are highly susceptible to quantum fluctuations. This aspect introduces additional complexities in understanding the behavior of electron glasses [8].*"
2. The sentence "*We attribute this discrepancy to the presence of slight oxygen vacancies, even in the pristine, as-received crystal*" in the caption of Fig. 1b has been modified to "*We attribute this difference to the difference in the sample preparation process, which may add slight oxygen vacancies, even in the pristine, as-received crystals (J. Phys.: Condens. Matter 6 4077 (1994)).*"
3. The sentence "*For details of the calculation we refer to the Supplementary Information Section 1.*" in caption of Fig. 1d has been modified to "*For details of the calculations, we refer to Supplementary Information Section 1 which significantly overlaps with our earlier work [Advanced Quantum Technologies 3, 2000021 (2020)].*"